# New Drug Targets to Prevent Death Due to Stroke: A Review Based on Results of Protein-Protein Interaction Network, Enrichment, and Annotation Analyses

**DOI:** 10.3390/ijms222212108

**Published:** 2021-11-09

**Authors:** Michael Maes, Nikita G. Nikiforov, Kitiporn Plaimas, Apichat Suratanee, Daniela Frizon Alfieri, Edna Maria Vissoci Reiche

**Affiliations:** 1Department of Psychiatry, Faculty of Medicine, Chulalongkorn University, Bangkok 10330, Thailand; 2Department of Psychiatry, Medical University of Plovdiv, 4002 Plovdiv, Bulgaria; 3IMPACT Strategic Research Center, Deakin University, Geelong, VIC 3220, Australia; 4Laboratory of Angiopathology, Institute of General Pathology and Pathophysiology, 125315 Moscow, Russia; nikiforov.mipt@gmail.com; 5National Medical Research Center of Cardiology, Institute of Experimental Cardiology, 121552 Moscow, Russia; 6Center for Precision Genome Editing and Genetic Technologies for Biomedicine, Institute of Gene Biology, 119334 Moscow, Russia; 7Advanced Virtual and Intelligent Computing (AVIC) Center, Department of Mathematics and Computer Science, Faculty of Science, Chulalongkorn University, Bangkok 10330, Thailand; kplaimas@gmail.com; 8Department of Mathematics, Faculty of Applied Science, King Mongkut’s University of Technology North Bangkok, Bangkok 10800, Thailand; apichat.s@sci.kmutnb.ac.th; 9Laboratory of Research in Applied Immunology, Health Sciences Center, State University of Londrina, Londrina, Paraná 86057-970, Brazil; frizon.alfieri@gmail.com (D.F.A.); reiche@sercomtel.com.br (E.M.V.R.)

**Keywords:** stroke, inflammation, neuroimmune, cytokines, hemostasis, coagulation, protein-protein interactions

## Abstract

This study used established biomarkers of death from ischemic stroke (IS) versus stroke survival to perform network, enrichment, and annotation analyses. Protein-protein interaction (PPI) network analysis revealed that the backbone of the highly connective network of IS death consisted of *IL6*, *ALB*, *TNF*, *SERPINE1*, *VWF*, *VCAM1*, *TGFB1*, and *SELE*. Cluster analysis revealed immune and hemostasis subnetworks, which were strongly interconnected through the major switches *ALB* and *VWF*. Enrichment analysis revealed that the PPI immune subnetwork of death due to IS was highly associated with *TLR2/4*, *TNF*, *JAK-STAT*, *NOD*, *IL10*, *IL13*, *IL4*, and *TGF-β1/SMAD* pathways. The top biological and molecular functions and pathways enriched in the hemostasis network of death due to IS were platelet degranulation and activation, the intrinsic pathway of fibrin clot formation, the urokinase-type plasminogen activator pathway, post-translational protein phosphorylation, integrin cell-surface interactions, and the proteoglycan-integrin extracellular matrix complex (ECM). Regulation Explorer analysis of transcriptional factors shows: (a) that *NFKB1*, *RELA* and *SP1* were the major regulating actors of the PPI network; and (b) hsa-mir-26-5p and hsa-16-5p were the major regulating microRNA actors. In conclusion, prevention of death due to IS should consider that current IS treatments may be improved by targeting VWF, the proteoglycan-integrin-ECM complex, TGF-β1/SMAD, NF-κB/RELA and SP1.

## 1. Introduction 

Acute ischemic stroke (IS) is one of the most prevalent major health problems worldwide, which frequently causes severe functional disabilities and mortality [1,2]. IS risk factors comprise unmodifiable factors including age, sex and ethnicity, and modifiable factors including diabetes mellitus, increased systolic blood pressure (≥140 mm Hg), increased body mass index or body weight, heart diseases, atrial fibrillation, transient ischemic attack (TIA), metabolic syndrome, smoking, sedentary life style, alcohol dependence, genetics and nutritional factors [1,2,3,4]. IS the consequence of blood vessel clots in the brain which interrupt blood flow and lead to lack of oxygen, causing cellular damage and neuronal cell death with neurodegenerative processes [5]. When vessels are occluded, inflammatory mediators are locally generated and propagated throughout the brain and peripheral, circulating blood, leading to neuroinflammatory processes and a systemic immune-inflammatory response [6].

Apart from being a neurodegenerative and neuroinflammatory disorder, IS also an atherothrombotic disease with activation of hemostasis, coagulation, fibrin clotting, and fibrinolysis cascades coupled with endothelial dysfunctions [7,8]. Thus, reduced levels of natural anticoagulants including protein C (PROC), protein S (PROS1), and antithrombin (SERPINC1), are not only associated with an increased risk of venous thrombosis, but also with the progression, outcome and prognosis of IS [8,9,10,11,12]. Major adhesion molecules, including vascular cellular adhesion molecule-1 (VCAM-1) and E-selectin (endothelial-leukocyte adhesion molecule 1/SELE) facilitate migration of leukocytes towards inflammatory regions and play a role in the outcome of IS [8].

The mean carotid intima media thickness (cIMT), as measured through carotid Doppler ultrasonography, is a marker of severity of atherosclerosis and is associated with the severity and outcome of IS [8,13]. The baseline IS severity and IS-induced disabilities may be assessed using the National Institutes of Health Stroke Scale (NIHSS) and the modified Rankin score (mRS) [14,15], respectively. Both scales are also useful to predict the functional outcome of IS, in terms of either short-term (three months) or long-term (one year) outcome [16,17,18,19].

IS the second leading cause of mortality, and around one-third of IS patients may die within the first few months after the ischemic event [20,21]. The cumulative risk for death at 28 days after IS (labeled as “28 days fatality” or “deaths from stroke”) is around 28%, and the risks at 1 and 5 years are 41% and 60%, respectively [22,23,24]. In nonfatal IS, the risk of death within the first year after the index stroke is 5 times higher than in a nonstroke population. In patients with initially nonfatal IS, the subsequent increased mortality rate is ascribed to vascular diseases, especially cerebrovascular and ischemic heart disease, neoplasms, infectious disease, diseases of the respiratory system, accidents and suicide [23].

Risk factors of increased mortality due to IS comprise increased age, increased cIMT scores, previous stroke and TIA, diabetes mellitus, heart disease, atrial fibrillation, increased baseline NIHSS and mRS scores, fever, and size of lesion [8,13,25,26,27,28,29]. Post IS death (three months and one year after admission) is predicted (in comparison with surviving IS patients) by different baseline blood-based biomarkers: (1) increased levels of glucose, but reduced levels of high-density lipoprotein (HDL) cholesterol and 25-hydroxyvitamin D [25(OH)D]; (2) reduced levels of the natural anticoagulants PROC, PROS1, and SERPINC1, and increased levels of Von Willebrand Factor (VWF), fibrinogen (FBG), and Factor 8 (F8); (3) increased levels of immune-inflammatory factors including white blood cell counts (WBC), interleukin (IL)-6, IL-10, tumor necrosis factor (TNF)-α, high sensitivity C-reactive protein (hsCRP), and ferritin; and lowered levels of transforming growth factor (TGF)-β1 and albumin (ALB); and (4) increased levels of the adhesion molecules SELE and VCAM-1 [8]. In fact, in machine learning models, different combinations of increased cIMT and NIHSS scores and these mentioned biomarkers yield the most accurate prediction of short- and long-term mortality due to IS [8,13].

A review of the existing literature shows that a number of genes may increase risk of mortality due to IS, including *SERPINE1* (serine protein inhibitor E1) or plasminogen activator inhibitor 1 (PAI-1), *ITGB3* (integrin beta-3), *MPG* (DNA-3-methyladenine glycosylase), *PROCR* gene (soluble endothelial protein C receptor), *HABP2* (Hyaluronan Binding Protein 2), *COL3A1* (type III collagen), *FBG*, *RETN* (resistin), *VKORC1* (Vitamin K epoxide reductase complex subunit 1), *INADL* (InaD-Like Protein, PALS1-Associated Tight Junction Protein), and *FXIIIA* Val34Leu genetic variant [30,31,32,33,34,35,36,37,38,39,40] Moreover, the micro-RNA (miR)-*10a rs3809783 A>T* (MIR10A) and *miR-34b/c rs4938723 T>C* genetic variant (MIR34B) and the long noncoding RNA (lncRNA) AL110200 are associated with increased mortality rates due to IS [41,42]. 

Although risk factor control, antithrombotic treatment, and revascularization have been widely applied in clinical practice, IS still a main cause of death [43]. The state-of-the-art management of IS comprises treatments targeting stroke-induced neurological damage and restoring the blood flow to the brain, including treatment with recombinant tissue plasminogen activator (tPA) IV [2]. Surgical decompression reduces risk of mortality, although in elderly patients, decompression may be accompanied by an increased risk of long-term dependency [44]. The Action Plan for Stroke in Europe considers that considerable effort should be focused on neuroimmune, neuroprotective, and vascular pathways [45].

Therefore, in the era of precision medicine, novel strategies for the prevention and treatment of IS are urgently needed [46]. Knowledge-based approaches, such as network, enrichment, and annotation analysis may provide insights into how genes and proteins interact to influence disease susceptibility. Moreover, protein-protein interaction (PPI) analysis may disclose new drug targets to develop novel therapies that modulate pathways and transcriptional factors thereby reducing death due to IS. Hence, the present study was conducted to delineate the characteristics of the PPI network of death due to IS versus stroke survival and to discover new putative drug targets in the pathways, cellular components, molecular patterns, and transcriptional factors enriched in the PPI networks of IS-associated mortality based on differentially expressed proteins (DEPs)/genes delineated in ours and other studies. 

## 2. Results

### 2.1. DEP Analysis

#### 2.1.1. The DEP PPI Network Topography of Death Due to Ischemic Stroke

Figure 1 displays the first-order protein network of death due to IS and it consists of 65 nodes with 806 edges, which exceeds the expected number of edges (n = 220) with a *p*-enrichment value of 1.0 × 10^−16^. The average node degree of this network is 24.8 with an average local clustering coefficient of 0.735. The network diameter = 3 and radius = 2, with a characteristic path length = 1.658, network density = 0.388 and heterogeneity = 0.445. The top six hubs were, in descending order of importance: IL6 (degree = 51), TNF (46), ALB (46), VCAM1 (38), IL10 (36) and VWF (35). The top two non-hub bottlenecks were: SELE (0.0298) and TGFB1 (0.0243). As such, the backbone of the network comprises eight proteins, including IL6, TNF, ALB, VCAM-1, IL-10, VWF, SELE, and TGF-β1.

MCL cluster analysis with an inflation parameter of three showed two protein communalities (see Figure 1): (a) a first cluster centered around immune DEPs and comprised TNF, IL6, IL10, VCAM1, ALB, CRP, SELE, TGFB1, and ferritin (FTH1); and (b) a second cluster centered around hemostasis DEPs including VWF, PROS, PROC, SERPINC1, and F8. Additionally, we identified the major switches between the two clusters as the top two DEPs that have the greatest number of connections (confidence level > 0.4) with the seed proteins in both clusters. As such, we found that the major switches between both clusters were ALB and VWF, which showed interconnections with all seed DEPs assigned to the other cluster (except VWF with FTH1). ALB, which belongs to cluster one, is interconnected with all cluster-one and cluster-two seed genes at a > 0.7 confidence level (except with FTH1 = 0.577), and shows interconnections at a confidence level > 0.9 with cluster-one (TGF-β1, CRP, IL-6) and cluster-two (PROC, VWF, PROS1, F8, FGB, SERPINC1) DEPs. VWF, which belongs to cluster two, is interconnected with all cluster-one and cluster-two DEPs at a confidence level > 0.4 (except with FTH1), while VWF shows interconnections at a confidence level > 0.7 with cluster one (ALB, CRP, SELE, TGF-β1, and VCAM-1) and cluster two (F8, FGB, PROC, PROS1, SERPINC1). In the first-order non-seed genes, we observed that vascular endothelial growth factor (VEGFA), which belongs to cluster one, was another switch between both clusters and was interconnected with all seed genes in cluster one (at >0.839) and with four cluster two seed DEPs, namely F8 (0.938), FGB (0.907), PROS1 (0.918), and SERPINC1 (0.447).

#### 2.1.2. Enrichment Analysis in the DEPs of Death Due to Ischemic Stroke

Table 1 shows the results of MCODE analysis using GO biological and molecular, KEGG, WikiPaths, PANTHER, and REACTOME protein sets, performed on the first-order genes. We observed a first complex representing an inflammatory response (MCODE1), a second reflecting formation of fibrin clot (MCODE2), and a third representing hemostasis including blood coagulation (MCODE3). Figure 2 shows a bar graph (heatmap) with the top 20 terms that were over-represented in the first-order network of death due to IS. Apart from the paths cited above, the following terms were over-represented, in descending order of importance: TNF signaling pathway, receptor signaling pathway via the Janus kinase-signal transducer and activator of transcription (JAK-STAT), extrinsic apoptotic signaling pathway, and the advanced glycation end products and their receptors (AGE-RAGE) signaling pathway in diabetic complications.

#### 2.1.3. Enrichment Analysis on Cluster One Genes of Death Due to Ischemic Stroke

Table 2 shows the REACTOME paths that were over-represented in the first-order network of cluster one genes. The most important REACTOME paths over-represented (and not shown in Figure 2 or Table 2) in cluster one were TNF receptor (TNFR)1-induced nuclear factor (NF)-κB signaling pathway, IL-10, IL-4 and IL-13 signaling, TNF-α signaling, IL-6 signaling, IL-1 family signaling, Death Receptor Signaling, and the Toll-Like Receptor 3 (TLR3) Cascade.

Figure 3 shows the heatmap of the (top 10) KEGG pathways that were enriched in cluster one, indicating that the most important paths over-represented in the network were the TNF-α and IL-17 signaling pathway, and lipid and atherosclerosis, TLR, and NF-κB signaling pathways. Figure 4 shows the top 10 InterPro domains which were statistically over-represented in the cluster one network, namely the chemokine IL18-like domain, death domain, type I cytokine receptor, CXC chemokine domain and TNFR/nerve growth factor receptor (NGFR) cysteine-rich region. We also performed GO function enrichment analysis on this first network, but the results do not provide any extra information other than that obtained using MCODE on all DEPs and REACTOME and KEGG pathways on the first DEP cluster.

#### 2.1.4. Enrichment Analysis on Cluster Two Genes of Death Due to Ischemic Stroke

MCODE analysis using KEGG, WikiPaths, PANTHER, REACTOME, and GO molecular and biological terms performed on the cluster two genes showed a first complex representing formation of fibrin clot, a second reflecting the urokinase-type plasminogen activator (uPA) and uPA receptor (uPAR)-mediated (uPA-uPAR) signaling pathway, elastic fiber formation and response to wounding (MCODE2) and a third representing reverse cholesterol transport, plasma lipoprotein particle organization, and protein-lipid complex subunit organization (MCODE3). Table 1 shows the features of both MCODE2 and MCODE3. We also performed KEGG enrichment analysis for this second protein network, but the results do not provide any extra information other than that presented in the MCODE analysis. 

Table 2 also shows the top 10 REACTOME paths enriched in the first-order cluster two network. Besides the hemostasis and fibrin clot terms listed above, this table shows that post-translational protein phosphorylation, integrin cell-surface interactions, the intrinsic pathway of fibrin clot formation, platelet degranulation and activation, signaling and aggregation, and a response to elevated platelet cytosolic Ca^2+^ were the most significant paths represented in the cluster two network. 

Figure 5 shows the top 10 GO molecular terms that were enriched in cluster two, indicating that (type 1) TGF-β1 receptor binding, endopeptidase inhibitor activity, protease binding, serine-type endopeptidase inhibitor activity, endopeptidase regulator activity, lipoprotein particle receptor binding, and inhibitory SMAD (I-SMAD) binding were the most important molecular functions. Figure 6 displays the top 10 InterPro domains which were statistically over-represented in the first-order cluster two network, namely integrin beta N-terminal and subunit, gamma-carboxyglutamic acid-rich (GLA) domain, MAD homology, and SMAD and epidermal growth factor (EGF)-like domains. 

### 2.2. DEP + Gene Analysis

#### 2.2.1. The DEP/Gene PPI Network Topography of Death Due to Ischemic Stroke

Figure 7 shows the first-order DEP/gene network of death due to IS. INADL was the only singleton in this network. This network contains 74 nodes and 881 edges which exceeds the expected number of edges (n = 227) with a p-enrichment value of 1.0 ×10^−16^. The average node degree of this network is 23.8 with an average local clustering coefficient of 0.705. The network diameter = 3 and radius = 2, with a characteristic path length = 1.728, network density = 0.344 and heterogeneity = 0.482. The top six hubs were in descending order of importance: IL6 (degree = 55), ALB (49), TNF (48), SERPINE1 (43), VWF (40) and VCAM1 (39). The same DEPs/genes were the top five bottlenecks, and the first two non-hub bottlenecks were TGFB1 (0.022) and SELE (0.013). Therefore, the backbone of this network comprises eight DEPs/DEGs namely IL6, ALB, TNF, SERPINE1, VWF, VCAM1, TGFB1, and SELE.

Using MCL cluster analysis with an inflation parameter of 3.4 we found two communalities as shown in Figure 7: (a) a first cluster centered around hemostasis-fibrin clot DEPs/genes including VWF, PROS, PROC, SERPINC1, F8, ALB, SERPINE1, ITGB3, MPG, PROCR, HABP2, COL3A1, FGB, and VKORC1; and (b) a second cluster centered around immune DEPs/genes including TNF, IL6, IL10, VCAM1, CRP, SELE, TGFB1, FTH1, and RETN. The major switches in this network were again ALB and VWF, which showed 17 and 21 interconnections at the >0.9 confidence level with cluster 1 and cluster 2 genes. In the first-order non-seed genes, we again found that VEGFA (belonging to cluster 2) was another switch between both clusters with 19 interactions (at >0.9) with genes in both clusters.

#### 2.2.2. Enrichment/Annotation Analysis on Cluster One DEPs/Genes of Death Due to Ischemic Stroke

Table 3 shows the results of MCODE analysis using KEGG, WikiPaths, PANTHER, REACTOME, and GO molecular and biological terms performed on the cluster one DEPs/genes, with a first complex showing hemolysis (not shown in Table 3), and a second complex (MCODE2) containing extracellular matrix (ECM) proteoglycans, focal adhesion and ECM organization.

Table 4 shows the top 10 KEGG-pathway-related enriched terms in the first DEP/gene cluster, indicating that the most important over-represented paths were complement and coagulation cascades, focal adhesion, proteoglycans, and the PI3K-Akt signaling pathway. Electronic Appendix A shows a heatmap (top 16) of the GO_TTRUST transcriptional factors that were enriched in cluster one (using Metascape), showing that “regulated by” SP1 and NFKB1 were the top transcription factor targets. Appendix A shows the enriched ontology domains in cluster one.

#### 2.2.3. Enrichment/Annotation Analysis on Cluster Two DEPs/Genes of Death Due to Stroke

Table 3 also shows the results of MCODE analysis using KEGG, WikiPaths, PANTHER, REACTOME, and GO molecular and biological terms performed on the cluster two DEPs/genes. This analysis showed one molecular complex (MCODE1) which accumulates TNF-α and IL-17 signaling pathways. Table 4 shows the top 10 KEGG pathways enriched in the second DEP/gene cluster. The most over-represented terms were TNF-α, TLR, IL-17, NF-κB, NOD-like receptor, and JAK-STAT signaling pathways. Appendix A shows a bar graph with the top 16 transcriptional factors (TTRUST) that were enriched in cluster two, indicating that ”regulated by” RELA and NFKB1 were the top transcription factor targets. Appendix A shows the enriched ontology domains in cluster two. Appendix A shows a heatmap with the top TTRUST factors that were enriched in all DEPs/genes, indicating that the top transcription factor targets were regulated by RELA and NFKB1. Appendix A shows the enriched ontology domains in all DEPs/genes combined. 

inBio Discover was employed to delineate which DOID diseases are over-represented in the DEPs/gene network. Table 5 displays the top 10 DOID annotations showing that the DEPs/genes are enriched in cardiovascular disease, blood coagulation disease, thrombosis, and immune and autoimmune disorders. Appendix A shows the extended network constructed with inBio Discover and some of the top DOID annotations. Appendix A showed the enriched GO terms of cellular components, including plasma membrane, blood microparticles, platelet alpha granules and the endoplasmic reticulum lumen. 

#### 2.2.4. Building a Composite Network with MultiOmics Enrichment Analysis

Appendix A shows a MultiOmics network (using InAct, mirNET, TTRUST and KEGG metabolic reactions) built using DEPs, genes, and miRNA (metabolic markers are not significant) in OmicsNet. The network comprises 1000 nodes and 2180 edges. Table 6 shows the results of OmicsNet enrichment analysis performed on the networks shown in Appendix A. The REACTOME pathways and PANTHER biological process enrichment analyses showed that the most important paths revolved around the metabolism of RNA, mRNA, RNA splicing, and nonsense-mediated dsecay and nonsense-mediated decay enhanced by the exon junction complex. A search for regulatory relationships using TTRUST shows that the top three over-represented transcriptional networks in the network were SP1 (82 hits), NFKB1 (45), and RELA (43). For example, NFKB1 regulates seed DEPs in the immune (CRP, IL6, TNF, TGFB1, SELE, VCAM1) and hemostasis (VWF, F8 and SERPINE1) subnetworks. Appendix A shows the targeted integration of these three regulating actors in the PPI network (shown in green colors).

Finally, we built another MultiOmics network (Appendix A) using all DEPs and genes and employed the Regulation Explorer to detect miRNA-gene interactions. This analysis shows that hsa-mir-16-5p (174 hits) and hsa-mir-26b-5p (170) were by far the most important interacting miRNAs. Appendix A shows the targeted integration of these two miRNAs in the network.

## 3. Discussion

### 3.1. The Networks and Subnetworks of Death Due to Stroke

The first major finding of this study is that the PPIs network of DEPs and DEPs/genes of death from IS versus stroke survival show a high connectivity and two interconnected subnetworks, namely, a first which is centered around immune genes and a second which is centered around hemostasis-coagulation genes. All query DEPs/genes participated in these networks except INADL. The backbone of this network consists of DEPs/genes which contribute to both subnetworks, namely IL6, TNF, TGFB1, VCAM, and SELE as authorities in the immune subnetwork, and ALB, SERPINE1, and VWF in the hemostasis subnetwork. Moreover, ALB ad VWF are important switches that connect both subnetworks because these DEPs show many significant interactions with proteins in both communities. Therefore, it appears that death due to IS predicted by an integrated response in a network which comprises strongly interconnected immune and hemostasis subnetworks. This indicates that both the immune response (as indicated by increased IL-6, IL-10, TNF-α, VCAM-1, SELE, and CRP but lowered TGF-β1 and ALB) and the activated hemostasis, thrombosis and coagulation pathways (as indicated by increased F8, VWF, and FBG but lowered PROC, PROS1, SERPINC1 and ALB levels) are intertwined phenomena leading to death from IS. The strong interconnections among the immune and hemostasis subnetworks have probably an evolutionary origin [47]. We will now discuss both the significant biological functions, paths, molecular complexes, and transcriptional factors enriched in the networks.

### 3.2. Terms Over-Represented in the Immune Subnetwork

The second major finding of this study is that the most significant paths and functions enriched in the networks of death due to IS were the inflammatory and NF-κB signaling pathways, NFKB1/RELA transcriptional factors, TNF-α and IL-17 signaling, and the TLR, TGF-β1, nucleotide-binding and oligomerization domain (NOD) and JAK-STAT pathways. These results indicate that altered expression of these pathways, and transcriptional factors, may lead to death due to IS.

There is now evidence that in IS, NF-κB may be a key factor for the transcriptional induction of cellular adhesion molecules, including SELE and VCAM-1, proinflammatory cytokines, including IL-6 and TNF-α, CRP, and FTH1, growth factors including TGF-β1, and coagulation factors including F8, PAI1, uPA, tissue factor and fibronectin [47,48]. Transient focal cerebral ischemia is accompanied by NF-κB activation in the nuclei of striatal and cortical neurons in the ischemic hemisphere [49]. It is important to note that ischemia-induced astroglial NF-κB may have neurodegenerative effects, whereas constitutive neuronal nuclear factor kappa B subunit 1 (NFΚB1) may have neuroprotective effects [50]. 

In animal models of middle cerebral artery occlusion, activation of astroglial NF-κB is downstream of TLR2 activation and a deficiency in TLR2/4 reduces the neurodeficit and IS size [50]. Moreover, models of focal cerebral ischemia show that increased NF-κB is associated with IS severity and size, and that there may be a causative association between both factors, whereby induced reductions in NF-κB levels are accompanied with lowered IS size and neurodeficit [48,51]. Inhibition of NF-κB (through administration of caffeic acid phenethyl ester, mycophenolate, atorvastatin, and cephalexin may attenuate oxidative stress and neurodegeneration due to middle cerebral artery occlusion [52]. Moreover, the promoter variant (rs11940017, -1727 C>T) of the *NFKB1* gene may affect IS susceptibility in the Korean population [53]. 

Furthermore, there is evidence that in IS the deleterious effects of increased NF-κB are associated with RELA activation [48], which is the second transcription factor which is highly significantly enriched in our PPI network. RELA plays a key role in the activation of NF-κB and the translocation of the latter to the nucleoplasm, while the RELA-NFKB1 complex mediates gene expression of many cytokines (UniPro UniProtKB—Q04206 (TF65_HUMAN) (uniprot.org (accessed on 19 September 2021). Moreover, the apoptotic responses caused by NF-κB are dictated by acetylation of RELA in Lys310, and RELA, but not p50, knockouts show a decreased infarct size [54,55]. 

The TLR2/4 complexes play a key role in IS with increased expression of both receptors being associated with the inflammatory response and progression of infarct volume and ischemic damage [56]. Both TLR levels (as assessed in peripheral blood monocytes using flow cytometry) are associated with increased plasma levels of IL-6, TNF-α, VCAM-1, and the clinical outcome and lesion volume [57]. It is interesting to note that in IS patients, serum levels of fibronectin, heat shock protein (HSP)60 and HSP70 are endogenous ligands leading to TLR2/4 activation [57]. Likewise, the inflammatory response during stroke is attenuated by blockade of the TLR2/4 complexes and cellular fibronectin. In TLR4 knockout mice, smaller stroke sizes and better neurocognitive functions are observed [58,59]. All TLR pathways activate NF-κB [56,60], indicating that activation of the TLR2/4 pathways and NF-κB signaling are interrelated phenomena. This suggests that the cumulative effects of both TLR2/4 and NF-κB signaling pathways may confer risk towards death due to IS. There are also reports that TLR4 genetic variants (e.g., *TLR4-119A* allele) are associated with an increased risk of IS [61]. Moreover, it is important to note that vitamin D3, which in our studies was associated with death due to IS, attenuates TLR2/4 expression and signaling, thereby preventing the translocation of p65 to the nucleus, and consequently, attenuating inflammatory responses [56]. 

Soon after IS onset, microglia cells and circulating monocytes are activated and consequently, increased release of TNF-α and IL-6 may be detected in the brain, cerebrospinal fluid, and bloodstream [8]. Increased TNF-α levels in CSF and blood are significantly associated with stroke outcome assessments including the Barthel Index and Scandinavian Stroke Scale (SSS) [62]. The plasma levels of TNF-α, which are observed in IS patients, may impact neuronal function and viability, even leading to neuronal death [63]. A recent meta-analysis shows that TNF-α is increased in Asian and Caucasian IS patients (overall SMD = 0.65, 95% CI = 0.29, 1.01), and that the *TNF-α-308 G > A* (*rs1800629*) genetic variant is associated with increased risk of IS [64]. IS is accompanied by an increased expression of IL-17 and IL-17RC in the serum in association with increased IL-6 and granulocyte-monocyte colony-stimulating factor (GM-CSF) and granulocyte colony-stimulating factor (G-CSF) [65]. It is thought that the IL-17-IL-17R pathway plays a key role in secondary poststroke damage via (a) effects on neutrophil infiltration in cerebral parenchyma, thereby promoting damage and thrombosis; (b) effects on tight junctions with blood-brain-barrier (BBB) breakdown and induction of neuronal apoptosis, and (c) synergistic effects with IL-6 [66]. Moreover, a polymorphism of the *IL17RC* gene is associated with a poorer prognosis of IS [65]. 

There are now some reports that in experimental models of IS, the JAK2-STAT3 pathway is activated and contributes to neuronal damage [67,68,69,70]. Middle cerebral artery occlusion is accompanied by increased concentrations of TNF-α, high mobility group box B1 (HMGB1), and phosphorylated JAK2/JAK2 and STAT3/STAT3 in the brain. Blocking JAK2, STAT3 and the JAK2/STAT3 signaling pathway significantly reduces IS-associated inflammatory responses [71]. Additionally, Wang and coworkers reported that in an IS model attenuation of the JAK-STAT pathway reduces apoptosis in neuronal cells and loss of neurological functions [72]. Cytokines including IL-6 and IL-10 activate Janus kinases causing translocation of STAT to the nucleus which, in turn, leads to changes in expression of a great number of immune and wound-healing genes [73]. 

NOD-like receptors are a family of cytoplasmic receptors which sense bacterial motifs and danger signals (including uric acid and ATP) and trigger an innate immune response by activating NF-κB and mitogen-activated protein kinase (MAPK) [74,75]. In the middle cerebral artery-occlusion model, the expression of NOD2 is significantly elevated, and ablation of the NOD2 gene reduces stroke size and inflammation as indicated by lowered expression of NF-κB, MAPK, IL-6, and TNF-α [75].

Our REACTOME path classifications revealed that our DEP network was highly significantly associated with IL-10, IL-4, and IL-13 pathways. We discussed before that increased activity of the IL-10 pathway in nonsurvivors may be part of a compensatory immune-regulatory mechanism which counterbalances an overzealous inflammatory response [8]. Elevated levels of IL-10 are often associated with a better functional outcome. For example, subjects with reduced plasma IL-10 in the first hours after IS have an increased risk of developing neurological symptoms two days later [76]. Some IL10 genetic variants are associated with increased risk of IS, including the *IL-10 rs1800896* variant [77]. IL-4 also has anti-inflammatory effects and drives macrophages from a M1 proinflammatory phenotype to a M2 phenotype with homeostatic, repair and immune regulatory properties, explaining how IL-4 administration may improve recovery in a mouse stroke model [78]. *IL4* KO mice show a worsened neurological outcome and increased brain damage following cerebral ischemia [79]. In IS, intracerebral delivery of IL-13, an anti-inflammatory cytokine, drives macrophages and microglia into the alternative activation state [80]. Nevertheless, high levels of anti-inflammatory cytokines such as IL-10 may sometimes be accompanied by a less favorable outcome, including increased risk of infections [81]. 

One of the major bottlenecks in the PPI network is TGF-β. This cytokine is elevated in the brain following IS and may exert anti-inflammatory, antiapoptotic and neuroprotective effects, thereby improving repair mechanisms and favoring nerve regeneration [82,83,84]. There are now some publications showing that genetic *TGFB1* variants are associated with IS [85]. Moreover, the effects of TGF-β1 on multiple target genes (including ECM) are mediated via the SMAD signaling pathway [86,87], which was significantly enriched in the hemostasis subnetwork list, albeit SMAD/MAD did not make the top five. Importantly, the TGF-β/SMAD2/3 signaling pathway has neuroprotective properties, for example by elevating Bcl-2 and lowering caspase-3 expression and decreasing microglial activation via NF-κB inhibition [88,89]. 

In a rat model of cerebral ischemia/reperfusion, SMAD3 administration may downregulate inflammatory and proapoptotic genes, suggesting that the TGF-β/SMAD pathway is a possible drug target [88]. All in all, the lowered levels of TGF-β1 in IS nonsurvivors versus survivors may contribute to the pathways leading to death due to IS. 

VCAM-1, another hotspot of the backbone of the PPI network, plays a critical role in the inflammatory response following IS, for example through adhesion of leukocytes to endothelial cells and transendothelial migration via interactions with integrin subunits [90,91]. Due to a fenestrated endothelium, the choroid plexus is the entry site for patrolling lymphocytes—mostly CD4^+^ central memory T cells—in the healthy brain [92]. To promote leukocyte trafficking, the choroid plexus epithelium constitutively expresses intercellular adhesion molecule (ICAM)-1 and VCAM-1, which, together with the mucosal vascular addressin cell-adhesion molecule, are upregulated in stroke [49]. In IS, intracranial VCAM-1 levels are associated with infarct and edema size [93]. Cell-adhesion molecules (CAM) KO models show a reduced infarct size, and administration of anti-CAM antibodies may decrease infarct size [94]. Increased levels of VCAM-1 and IL-6 predict a new vascular incident, and thus may increase risk of death due to IS [95]. 

### 3.3. Terms and Functions Over-Represented in the Hemostasis Subnetwork 

The second most important subnetwork in death due to IS was centered around hemostasis, thrombosis and coagulation genes, which were enriched especially in fibrin clotting or the clotting cascade, TGF-β binding, the uPA and uPAR-mediated (PID-uPA-uPAR) signaling pathways, ECM proteoglycans, integrin cell-surface interactions and platelet activation, aggregation and degranulation. Although abnormalities in the hemostasis axis are not a common cause of IS [96,97,98], perturbations are seen following stroke onset, and these may aid as biomarkers in diagnosis and severity of IS and prediction of the treatment response. This is further substantiated by our findings that a response to wounding is over-represented in the hemostasis subnetwork. It could be argued that a combined deficiency in the PROC, PROS1 and SERPINC1 proteins in the hemostasis network may contribute to the response to wounding. Nevertheless, deficits in these proteins are not a common cause of IS [99], although our studies suggest that they may contribute to death due to IS. 

One major hotspot of the backbone of the PPII network and a major connector of both subnetworks is VWF, which shows many interactions with genes from both the immune and hemostasis networks. Such data agree with the view that VWF mediates the crosstalk between immune cells and hemostasis mechanisms and contributes to inflammation, including vascular inflammation [100,101]. The VWB factor is released during the rupture of the endothelial layer of the vessels, whereby the consequent exposure of collagen to platelets leads to clot formation [102]. Consequently, endothelial cells release VWF, P-selectin, SELE, and inflammatory mediators [103]. VWF promotes platelet adhesion to the damaged site by forming a molecular bridge between the subendothelial collagen matrix and the platelet surface receptor complex GPIb-IX-V [101]. VWF levels are increased in IS patients and are associated with the cardioembolic and large-vessel disease subtypes [104]. Additionally, VWF levels are associated with severity of arterial thrombus formation and poor functional outcomes [105,106]. Furthermore, VWF may function as a biomarker of the response to thrombolytic or endovascular treatment in IS patients [106,107]. Variations in the *VWF* gene (Sma I) may be associated with an increased risk of IS [108]. Preclinical and clinical studies using VWF antagonists and combining the latter with tPA could prevent microvascular thrombus formation, thereby attenuating the progression of IS [106]. Additionally, other authors proposed that targeting VWF-mediated platelet activation and adhesion is a new drug target to treat IS [109].

Fibrinogen (FBG) is one of the hemostasis biomarkers with an essential role in the thrombosis process because it is related to platelet aggregation after injury and inflammation [110,111]. FBG is released from the liver into the bloodstream and is cleaved by thrombin at the damaged site, resulting in fibrin formation. Fibrin is one of the main constituents of blood clots and provides remarkable biochemical and mechanical stability [111]. FBG and CRP levels can independently predict the risk of early death in middle-aged IS patients, emphasizing the role of inflammation and coagulation in the evolution of IS. For each 10 mg/dL increase in FBG levels there was a 18% higher risk of dying, while for 1 mg/L increase in CRP the additive risk was 18%. FBG levels >490 mg/dL and CRP levels >18 mg/L were the optimal points that discriminated those who died from survivors [112]. Another study showed that hyperfibrinogenemia, defined as a plasma FBG concentration >350 mg/dL, predicts the long-term risk of death in IS patients [113]. Moreover, FBG has been used to evaluate the long-term outcome and the size of the clot burden in patients after IS [114,115]. Furthermore, an increase in FBG levels is observed after IS and is associated with infarct size and outcome [116,117,118]. Elevated hemostatic markers after acute IS stroke identify patients with increased risk for mortality independently of IS severity or stroke type [119]. Higher concentrations of fibrinopeptide A, b-thromboglobulin, prothrombin fragments 1 and 2, thrombin-antithrombin complexes, platelet factor four and VWB factor have all been associated with a worse clinical course in IS and increased mortality [119,120,121]. In addition, patients with high PAI-1 levels are less likely to achieve recanalization after receiving tPA, and experience poorer outcomes [122].

Another possible major pathway which was found using enrichment analysis is the uPA-uPAR pathway. This pathway is activated in the periphery of growth cones of the injured neurons in IS brains, where it induces repair mechanisms [123]. The binding of uPA to its receptor promotes β1-integrin recruitment to the plasma membrane with consequent activation of “small Rho GTPase Rac1 and Rac1-induced axonal regeneration” [123]. Since this process is regulated by the low-density lipoprotein receptor (LRP1), it was proposed that the uPA-uPAR-LRP1 axis is another possible drug target in IS and by inference may be a new drug target to prevent death due to IS. Moreover, recombinant uPA may protect the integrity of pre- and post-synaptic terminals and astrocytic elongations against the detrimental effects of IS [124]. Unfortunately, uPA also increases the production of reactive oxygen species and NADH oxidases (Nox1 and Nox4) and enhances superoxide production by neutrophils, findings which suggest that reducing uPA functions may be beneficial [125].

Our enrichment and annotation analyses revealed that the hemostasis subnetwork genes were significantly associated with ECM proteoglycans and integrin cell surface interactions, which are components of the BBB. These functions are impacted by IS, the consequent vasogenic edema, early angiogenesis, inflammation, and reperfusion injury [126,127,128,129]. These IS-induced processes strongly impact the BBB, resulting in breakdown of the tight junctions and paracellular barrier and increased BBB permeability, which may further worsen vasogenic edema and neuroinflammation through increased entry of inflammatory mediators and activated T cells into the brain [130,131]. As observed in our paper and previously, such processes may lead to increased mortality due to IS. Moreover, proteoglycans, integrins and fibronectin are key ECM adhesion receptors which integrate external and internal signals, and regulate global cellular processes, cellular signaling, and cell growth, proliferation, migration, and survival [126]. The ECM regulates the tight junctions, neurons, astrocytes, and the vasculature and plays a key role in wound healing, cell homeostasis, and tissue and neuronal regeneration [132]. Importantly, the proteoglycan-integrin-ECM complex and related BBB functions including tight junctions are now considered to be new drug targets in the treatment of IS, for example by targeting integrins and MMPs [128,129].

### 3.4. Interactions, Pathways, and Functions Which Bridge the Immune and Hemostasis Subdomains

Our network analyses showed that not only the VWF (as discussed in the previous section) but also ALB is a major controller gene between both subnetworks of death due to IS and, consequently, that ALB may also be an important drug target. ALB has anti-inflammatory, antioxidant and neuroprotective capacities and thus plays a role in the immune subnetwork, although a second clustering analysis performed in our study allocated ALB into the hemostasis network. ALB also shows antiplatelet aggregability and is a carrier for two anticlotting compounds, namely heparin cofactor and SERPINC1 [133,134]. Subjects with low ALB display increased primary hemostasis, and enhanced platelet aggregation and clot formation, explaining how low ALB levels observed in nonsurvivor IS patients may increase the vulnerability to develop venous thromboembolism [133]. Importantly, reduced baseline ALB levels reflect a chronic inflammatory state, which is already present weeks before the IS (because the half-life of ALB is 21 days). All in all, lowered ALB levels may predispose patients towards interrelated aberrations in both the immune and hemostasis subnetworks, thereby, increasing risk of death due to IS.

In the enlarged giant network, VEGFA is another switch between the immune and hemostasis communalities. This gene is a member of the VEGF family and the protein acts as a mitogen, thereby activating endothelial cells and regulating neuronal cells [135]. VEGFA is implicated in the pathophysiology of atherosclerosis, and in stroke, VEGFA is elevated in the ischemic penumbra and mediates vessel and neuron repair and remote plasticity in ischemic brain regions [135]. Treatment with VEGFA coupled with stem cells may show therapeutic effects in animal stroke models. Moreover, there is an association between IS and different genetic variants of *VEGF* genes (e.g., *-2578C>A* and *936C>T* variants) [136].

There are many more pathways and functions which link inflammation and coagulation, including TLRs (impact coagulation, platelet activity and aggregation, and thrombosis) [137], NOD2 signaling (enhances platelet activation and thrombosis) [138], TNF-α signaling (activates coagulation, fibrinolysis, neutrophil degranulation, and the release of secretory phospholipase A2, predominantly mediated by the p55 TNFR) [139], platelets (platelet degranulation causes increased cIMT and thus increased inflammatory atherosclerotic processes) [47,140], protease-activated receptors and thrombin, complement, neutrophil extracellular traps, and microparticles [141]. 

Moreover, our TTRUST enrichment analysis shows that not only NF-κB and RELA (discussed in the previous sections), but also SP1 is a major regulator of both the immune and hemostasis subnetworks. NF-κB is not only critically involved in modulating inflammatory processes but also in thrombotic responses [47]. Thus, NF-κB activation increases the thrombogenic potential, activates platelets, and promotes coagulation, microthrombi formation, immunothrombosis and thromboinflammatory disease [47]. In addition, *F8*, *PAI1, uPA* and *F3* are target genes of NF-κB. SP1 or transcription factor SP1 (or specificity protein 1) regulates the expression of housekeeping genes which are involved in immune responses, apoptosis, chromatine remodelling, cell differentiation and cell growth [142]. This transcription factor regulates different subnetworks including the immune response (including TNF-α), MAPK and JAK-STAT pathways, platelet activation, ECM, and blood vessels [142]. 

Our REACTOME pathway and PANTHER process enrichment analyses performed on all markers including miRNA levels revealed the strong impact of translation, RNA, metabolism of mRNA, RNA splicing, and nonsense-mediated decay on the interactome of death due to IS, indicating that dysfunctions in postischemic translation regulation are involved. During ischemia and brain cell injury, translation may arrest due to lack of ATP, and changes in translational regulation at the mRNA level and the ribosomic network may develop, which may cause a multitude of aberrations in the downstream PPI and metabolic network modules [143,144]. The nonsense-mediated decay pathway degrades transcripts with a premature stop codon, thereby reducing errors in gene expression. Failures in this surveillance system may be accompanied by increased synthesis of abnormal, including toxic, proteins [145]. 

It is interesting to note that both miRNAs, which are altered in death due to IS, regulate immune functions, with hsa-mir-10a regulating IL-8, IL-6, TNF-α, GATA6 apoptotic pathways, and hsa-mir-34b regulating innate immunity (target mining in www.mirbase.org (accessed on 19 September 2021)). Our study also showed that the expression of the most important miRNA, which is over-represented in death due to IS (hsa-mir-16-5p), was found to be increased in IS and to regulate NF-κB transcription [146,147]. Moreover, also the second-most important miRNA (hsa-mir-26b-5p) is part of a long noncoding RNA (LncRNA)-miRNA-mRNA network of IS and inhibits NF-κB expression [148,149]. Some miRNAs are associated with IS risk factors including hypertension (miR-155), atherosclerosis (miR-21, miR-126, miR-143), atrial fibrillation (miR-26), diabetes mellitus (miR-124a, miR-126), and dyslipidemia (miR-33, miR-122), while some miRNA antagonists have the potential to act as neuroprotective molecules [150]. Elevated expression of miR-15a, miR-16, and miR-17-5p in the serum is strongly associated with IS [151]. 

## 4. Methods

### 4.1. Selection of Seed Proteins, Genes, miRNA, and Metabolic Markers

“This study is a secondary data analysis on existing data using open, deidentified and noncoded data sets and, therefore, this is nonhuman subjects research which is not subject to the Institutional Review Board (IRB) approval”. In our previous case-control studies, we identified DEPs and metabolic pathways in IS patients who died three months to one year after the IS event and compared them with IS survivors [8]. We found that the markers of death due to IS after one year were the same as those after three months and vice versa, and therefore, these biomarkers reflect pathways leading to death within one year after IS. The clinical features and sociodemographic data of the patients are shown elsewhere [8,13,152]. Appendix A shows the characteristics of those DEP study samples included in the current study. Importantly, none of the patients were treated with tPA, and blood for the assay of biomarkers was sampled within 24 h after admission into hospital. We were able to include 15 proteins which were significantly altered in nonsurvivors versus survivors following stroke, namely SERPINC1, PROC, PROS1, VWF, coagulation F8, FBG, CRP, ALB, TNF-α, IL-10, IL-6, TGF-β1, FTH1, VCAM-1, and SELE [8,13,152]. The downregulated DEPs were SERPINC1, PROC, PROS1, ALB, and TGF-β1, whereas the other DEPs were upregulated. In our studies, we also delineated three Kyoto Encyclopedia of Genes and Genomes (KEGG) metabolic pathways of death due to IS, namely C00103 (glucose), C05443 (vitamin D) and Hsa04979 (cholesterol) [8,13,152]. Moreover, we added a) a number of genes which are associated with death due to IS as reviewed in the Introduction, namely SERPINE1, ITGB3, MPG, PROCR, HABP2, COL3A1, FGB, VKORC1, INADL, and RETN; and b) two miRNAs which are associated with death due to IS, namely MIR10A (*miR-10a* rs3809783) and MIR34B (*miR-34b/c* rs4938723). Appendix A shows the features of those studies that were included in the current study. 

### 4.2. PPI Network Construction, and Enrichment and Annotation Analyses

We constructed three network types, namely a first based on the DEPs delineated in our clinical studies, a second on the combined DEPs + genes; and a third on all data combined. PPI networks were constructed with network expansion as explained previously [153]. In brief, IntAct Molecular Interaction Database (https://www.ebi.ac.uk/intact/ (accessed on 19 September 2021)), a primary database based on peer-reviewed publications, and STRING version 11.0 (String Consortium, 2021, https://string-db.org (accessed on 19 September 2021), a predictive database, were used to construct the networks. We constructed zero-order PPIs (seed proteins only) and first-order PPIs (50 interactions in the first shell, none in the second shell) (set organism: homo sapiens, and a minimum required interaction score of 0.400). The network characteristics (number of nodes and edges, etc.) were computed using STRING and the Cytoscape plugin Network Analyzer. The backbone of the network was delineated and consisted of the top hubs (nodes with the highest degree) and the top non-hub bottlenecks (nodes with the highest betweenness centrality). The physical interactions between the proteins/genes/mir were visualized using STRING, Cytoscape (https://cytoscape.org (accessed on 19 September 2021), Gene Ontology (GO) net (https://tools.dice-database.org/Gonet/ (accessed on 19 September 2021), OmicsNet (OmicsNet), inBio Discover (https://inbio-discover.com/ (accessed on 19 September 2021), and Metascape (https://metascape.org (accessed on 19 September 2021). Markov Clustering (MCL) was conducted using STRING to detect communalities of highly interconnected nodes with similar attributes and functions. Separate enrichment/annotation analyses were conducted on the formed communalities and upregulated and downregulated genes as well. 

The different PPI networks were examined for their enrichment scores and annotated terms employing (a) STRING to establish the GO biological processes and molecular functions, and KEGG (https://genome.jp/kegg/ (accessed on 19 September 2021) and REACTOME (the European Bio-Informatics Institute Pathway Database; https://reactome.org (accessed on 19 September 2021) pathways; (b) Enrichr, a gene list enrichment analysis tool (https:/maayanlab.cloud/Enrichr/ (accessed on 19 September 2021) to establish and visualize (via Appyters) GO molecular functions, KEGG pathways, and InterPro domains (InterPro (ebi.ac.uk (accessed on 19 September 2021 ); (c) OmicsNet (using InAct, mirNET, TTRUST and KEGG metabolic reactions) to establish REACTOME and PANTHER (PANTHER—Gene List Analysis (pantherdb.org (accessed on 19 September 2021) biological processes, and the OmicsNet regulation explorer to establish TTRUST transcriptional regulatory relationships (www.grnpedia.org/trrust (accessed on 19 September 2021); (d) Metascape to establish and visualize the over-represented REACTOME, KEGG, PANTHER, and Wiki (WikiPathways—WikiPathways) pathways; (e) inBio Discover to establish DOID human disease phenotypes (Disease Ontology—Institute for Genome Sciences @ University of Maryland (disease-ontology.org (accessed on 19 September 2021); and (f) R package ClusterProfiler to stablish the cellular components over-represented in the network. In addition, Metascape was combined with Molecular Complex Detection (MCODE) to delineate smaller molecular complexes and to visualize enriched ontology clusters. The latter are based on accumulative hypergeometric *p*-values which are used for filtering whereby the remaining terms are hierarchically clustered into a tree, which is based on a 0.3 kappa score threshold, and is casted into term clusters. OmicsNet was used to build a composite network (consisting of DEPs, genes, miRNA, and metabolic paths) coupled with multiomics enrichment analysis. We use false discovery rate (FDR)-corrected *p*-values.

## 5. Conclusions

This study defined (a) the pathways and molecular processes and ensuing pathoclinical conditions (blood coagulation and thrombosis, and immune system and cardiovascular system disease) leading to death due to IS; and (b) the possible drug targets which should be utilized to prevent death due to IS. Foremost, the major switches between the immune and hemostasis subnetworks (ALB and VWF) could be targeted to dampen the coordinated response in both subnetworks. Nevertheless, treatment with intravenous ALB did not improve IS outcome and may even cause pulmonary edema and intracerebral hemorrhage [154]. Targeting VWF, on the other hand, may be a more promising approach via reducing stroke recurrence, thrombotic risk, platelet aggregation and VWF-associated inflammation [155,156,157]. Improving remote plasticity in the ischemic hemisphere by targeting VEGF, promoting neuronal repair by targeting the TGF-β1/SMAD, and targeting the proteoglycan-integrin-ECM complex and related BBB functions are other possibilities which may reduce IS-induced neuronal damage and promote the repair of affected cells. Nevertheless, administration of VEGF and increased endogenous VEGF signaling may have negative consequences in acute stroke due to BBB breakdown causing neuroinflammation and intracranial pressure [158]. There are some studies which have shown that TGF-β1 has a neuroprotective effect against ischemia-associated neuronal injuries and excitotoxicity [159]. Preclinical research shows that some proteoglycans (mimetics or recombinant proteoglycan cores) may be attractive drugs for targeting (neuro)degenerative disorders, including stroke [160]. For example, in a rodent model, heparan sulfate proteoglycan 2 shows improved angiogenesis after IS and is neuroprotective [161]. 

Furthermore, the most influential genes of the network are other possible drug targets and comprise (besides ALB, VWF and TGFB1) IL6, TNF, VCAM1, IL10, and SELE. However, selectively targeting any of these backbone DEPs may improve only part of the network, although combinatorial treatment with other pathways may be more beneficial. Targeting the anti-inflammatory cytokines IL-10, IL-13, and IL-4, which were shown to reduce infarct size, is another option. In this respect, IL-10 would be a good candidate because this cytokine is part of the backbone of the PPI network, although in some conditions this cytokine may increase risk of infections.

Targeting the pathways and molecular complexes that are involved in death due to IS may provide better target options, because these pathways/processes govern many players in the PPI network. Candidates are the JAK-STAT, NOD, and TLR2/4 pathways. However, since all these pathways appear to play a role in death due to IS, a combinatorial treatment targeting some of these pathways appear to be indicated. Recently, Zhu et al. [162] detected that, in animal models, Janus Kinase inhibition may improve IS injury and neuroinflammation. Nevertheless, there are many contradictory results of targeting Janus Kinases in IS models, as reviewed by Raible et al. [163]. A recent review showed that targeting the NLRP3 inflammasome may offer another strategy to improve IS [164]. The TLRs, and especially TLR2/4, play a critical role in IS, the pathological development of IS brain damage, and reperfusion injury [56,165]. Inhibition of the TLR4-NOX4 pathway, a predominant causal pathway in IS, protects against oxidative stress and neuronal apoptosis [166]. In addition, preconditioning of the TLR response to damage-associated patterns may be associated with increased neuroprotection [167]. Alternatively, targeting the transcription factors (NFKB/RELA or SP1) which govern a large part of the proteome network may be an adequate strategy. Recently, Howell and Bidwell [168] reviewed the potential benefits of targeting NF-κB in ischemia-reperfusion injury in the brain. 

Finally, this study also observed that hsa-mir-16-5p and hsa-mir-26b-5p may be targeted or used as treatments. For example, hsa-mir-26b-5p is part of a LncRNA-miRNA-mRNA network of IS, and appears to regulate the transcriptome and proteome of death due to IS. Importantly, treatment of death due to IS should also target metabolic risk factors such as glucose, HDL-cholesterol, and 25(OH)D.

Additionally, none of the subjects in our DEP studies received tPA, which could have influenced biomarker levels and outcome data. As such, our DEP results are representative of the pathways leading to death in a cohort of IS patients who did not receive tPA treatment. In the second section of our studies, we included gene studies, and only one of these adjusted the outcome data for tPA treatment, while another was performed on stroke subjects who received tPA. Most studies, however, did not offer any information concerning tPA treatment. Therefore, future research should focus on the PPI network associated with death from IS in patients treated with tPA, in order to delineate drug targets that may augment the efficacy of tPA in preventing death from IS.

Our findings suggest that current treatments aimed at lowering post-IS mortality should target VWF, the proteoglycan-integrin-ECM complex, TGF-β1/SMAD, NFKB/RELA, and maybe SP1.

## Figures and Tables

**Figure 1 ijms-22-12108-f001:**
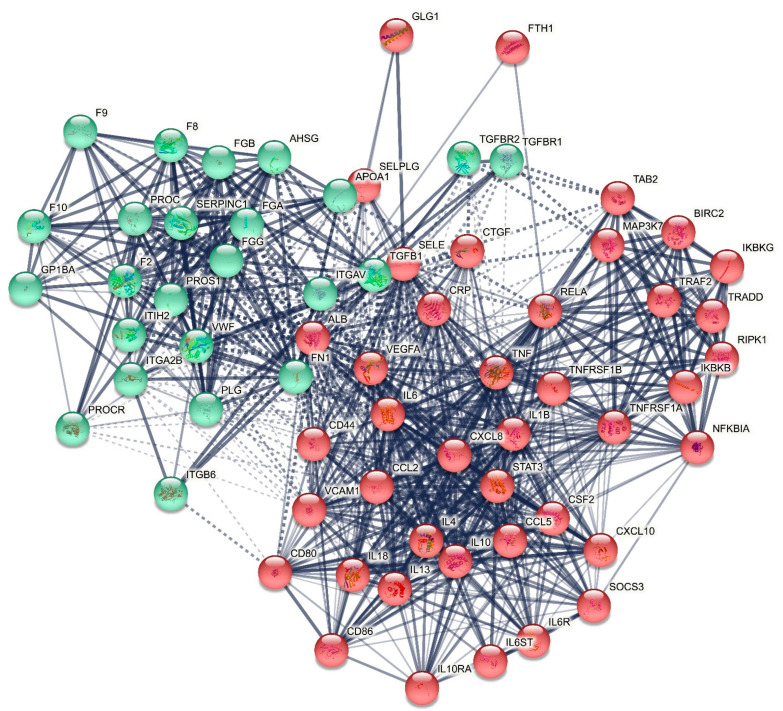
First-order protein network showing the results of Markov Clustering (MCL) analysis: (1) a first immune cluster (red bullet icons) was centered around ALB (albumin), CRP (C-reactive protein), IL10 (interleukin 10), IL6 (interleukin 6), SELE (E-selectin), TGFB1 (transforming growth factor beta 1), TNF (tumor necrosis factor), and VCAM1 (vascular cellular adhesion molecule 1), and (2) a second hemostasis fibrin-clot cluster (green bullet icons) centered around F8 (coagulation factor VIII), FGB (beta-fibrinogen), PROC (protein C), PROS1 (protein S), VWF (von Willebrand Factor), and SERPINC1 (antithrombin).

**Figure 2 ijms-22-12108-f002:**
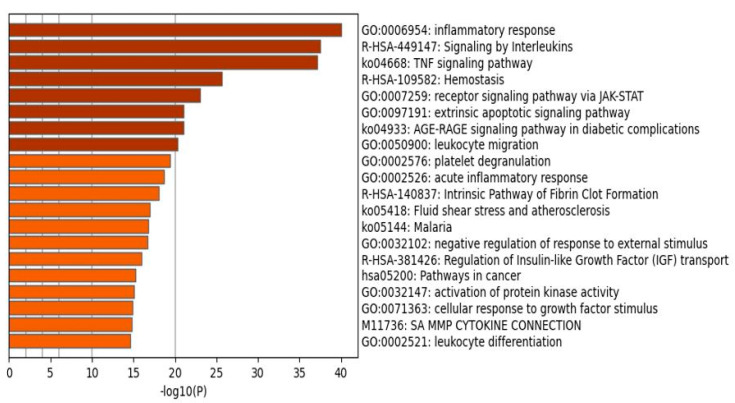
Heatmap of enriched terms showing the top 20 functions that were overexpressed in the network of patients who died due to ischemic stroke (accumulative hypergeometric *p*-values).

**Figure 3 ijms-22-12108-f003:**
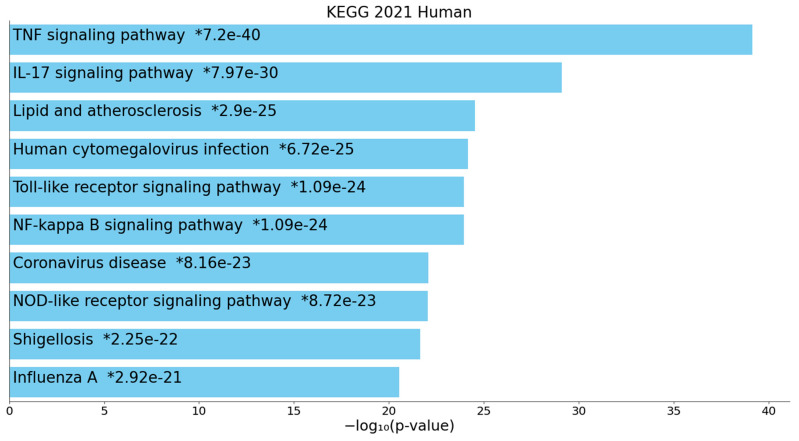
Heatmap of enriched KEGG (Kyoto Encyclopedia of Genes and Genomes) terms showing the top 10 functions that were overexpressed in the first protein subnetwork of patients who died due to ischemic stroke (accumulative hypergeometric *p*-values). TNF: tumor necrosis factor; IL: interleukin; NF-kappa B: nuclear factor kappa B; NOD: nucleotide-binding, and oligomerization domain. “Colored bars correspond to terms with significant *p*-values (<0.05); the * indicate that the term has a significant adjusted *p*-value (<0.05)” (Appyters, Appyter (maayanlab.cloud) as accessed on 6 November 2021).

**Figure 4 ijms-22-12108-f004:**
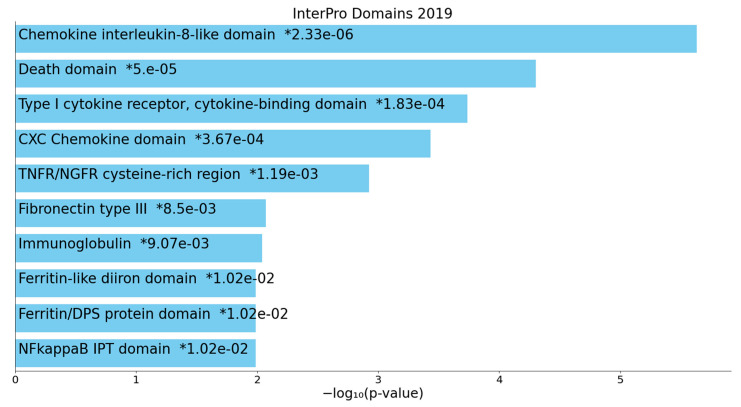
Heatmap of enriched InterPro domains showing the top 10 functions that were overexpressed in the first protein subnetwork of patients who died due to ischemic stroke (accumulative hypergeometric *p*-values). “Colored bars correspond to terms with significant *p*-values (<0.05); the * indicate that the term has a significant adjusted *p*-value (<0.05)” (Appyters, Appyter (maayanlab.cloud) as accessed on 6 November 2021).

**Figure 5 ijms-22-12108-f005:**
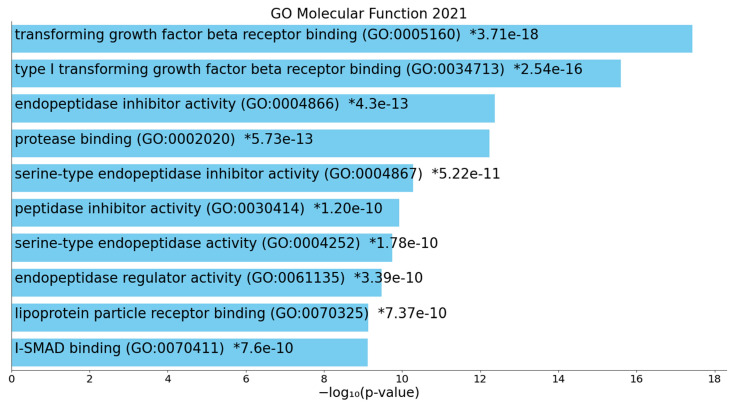
Heatmap of enriched Gene Ontology (GO) molecular functions that were overexpressed in the second protein subnetwork of patients who died due to ischemic stroke (accumulative hypergeometric *p*-values). I-SMAD: inhibitory SMAD. “Colored bars correspond to terms with significant *p*-values (<0.05); the * indicate that the term has a significant adjusted *p*-value (<0.05)” (Appyters, Appyter (maayanlab.cloud) as accessed on 6 November 2021).

**Figure 6 ijms-22-12108-f006:**
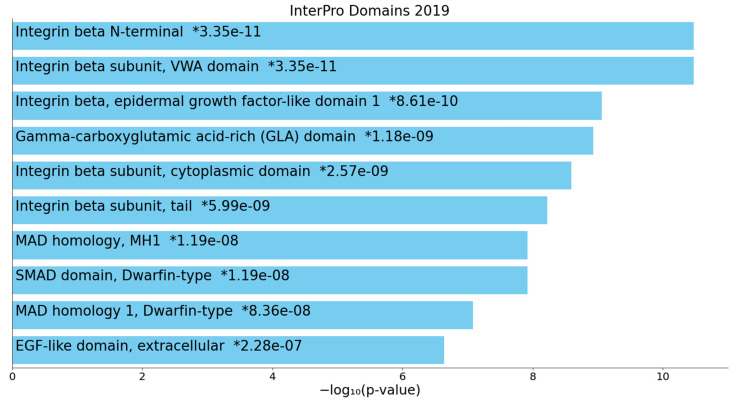
Heatmap of enriched InterPro domains showing the top 10 functions that were overexpressed in the second protein subnetwork of patients who died due to ischemic stroke (accumulative hypergeometric *p*-values). VWA: von Willebrand factor type A; MAD: Mothers against Dpp (decapentaplegic); MH1: MAD 1 homology (N terminus); SMAD: Superfamily of Mothers against Dpp; EGF: Epithelial growth factor. “Colored bars correspond to terms with significant *p*-values (<0.05); the * indicate that the term has a significant adjusted *p*-value (<0.05)” (Appyters, Appyter (maayanlab.cloud) as accessed on 6 November 2021).

**Figure 7 ijms-22-12108-f007:**
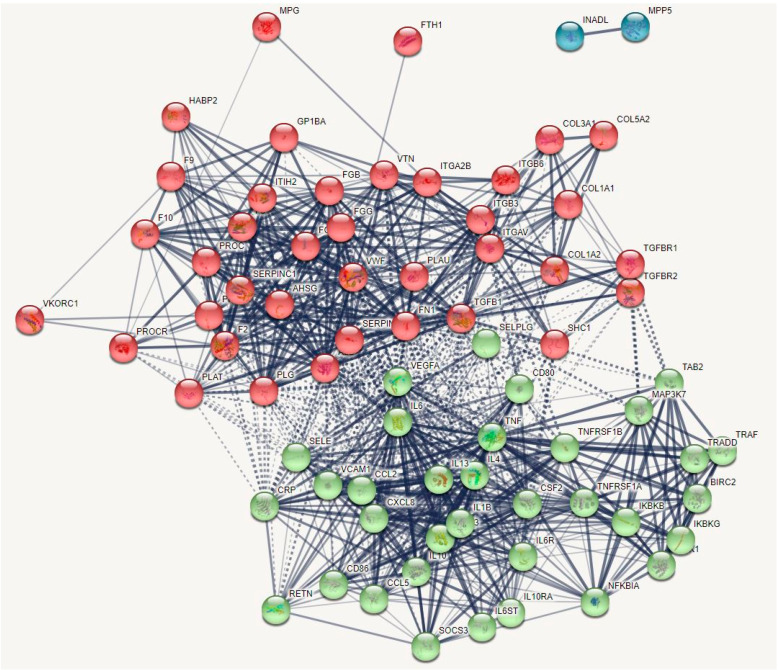
First-order protein network showing the results of Markov Clustering (MCL) analysis with (1) a first hemostasis fibrin-clot cluster (red bullet icons); and (2) a second immune cluster (green bullet icons). Blue bullets: not belonging to any cluster.

**Table 1 ijms-22-12108-t001:** Results of Molecular Complex Detection (MCODE) analysis performed on the differently expressed proteins (DEPs) of death due to ischemic stroke.

MCODE Components	GO ID	Biological Term	Log10 (*p*)
All DEPs, MCODE1	R-HSA-449147	Signaling by Interleukins	−42.1
	GO:0006954	Inflammatory response	−40.9
	ko04668	Cytokine signaling in immune system	−40.1
All DEPs, MCODE2	R-HSA-140877	Formation of fibrin clot (Clotting Cascade)	−19.4
	R-HSA-140875	Common pathway of fibrin clot formation	−17.5
	GO:0050819	Negative regulation of coagulation	−15.1
All DEPs, MCODE3	R-HSA-109582	Hemostasis	−6.6
	GO:0007596	Blood coagulation	−5.1
	GO:0007599	Hemostasis	−5.1
Cluster Two DEPs, MCODE2	M174	PID UPA UPAR PATHWAY	−21.5
	R-HSA-1566948	Elastic fiber formation	−20.2
	GO:0009611	Response to wounding	−20.2
Cluster Two DEPs, MCODE3	GO:0043691	Reverse cholesterol transport	−9.5
	GO:0071827	Plasma lipoprotein particle organization	−8.5
	GO:0071825	Protein-lipid complex subunit organization	−8.1

**Table 2 ijms-22-12108-t002:** REACTOME path classifications of the differently expressed proteins in cluster one of death due to stroke.

Path ID	Pathway Names in Cluster One	Found	Total	Ratio	*p* Value	pFDR
R-HSA-5357956	TNFR1-induced NF-kappa B signaling pathway	11	30	0.002063	1.11 × 10^−16^	6.77 × 10^−15^
R-HSA-6783783	Interleukin-10 signaling	31	86	0.005914	1.11 × 10^−16^	6.77 × 10^−15^
R-HSA-6785807	Interleukin-4 and interleukin-13 signaling	28	211	0.01451	1.11 × 10^−16^	6.77 × 10^−15^
R-HSA-449147	Signaling by interleukins	55	643	0.044217	1.11 × 10^−16^	6.77 × 10^−15^
R-HSA-1280215	Cytokine signaling in immune system	62	1092	0.075093	1.11 × 10^−16^	6.77 × 10^−15^
R-HSA-168256	Immune system	67	2681	0.184363	1.11 × 10^−16^	6.77 × 10^−15^
R-HSA-75893	TNF signaling	11	51	0.003507	1.67 × 10^−14^	8.66 × 10^−13^
R-HSA-1059683	Interleukin-6 signaling	8	17	0.001169	1.55 × 10^−13^	7.13 × 10^−12^
R-HSA-446652	Interleukin-1 family signaling	14	167	0.011484	1.23 × 10^−12^	5.04 × 10^−11^
R-HSA-5357905	Regulation of TNFR1 signaling	9	41	0.002819	3.82 × 10^−12^	1.38 × 10^−10^
R-HSA-73887	Death receptor signaling	13	158	0.010865	1.05 × 10^−11^	3.46 × 10^−10^
R-HSA-6783589	Interleukin-6 family signaling	8	30	0.002063	1.37 × 10^−11^	4.11 × 10^−10^
R-HSA-168164	Toll-Like Receptor 3 (TLR3) cascade	10	102	0.007014	5.59 × 10^−10^	1.56 × 10^−08^
R-HSA-937061	TRIF(TICAM1)-mediated TLR4 signaling	10	107	0.007358	8.81 × 10^−10^	2.11 × 10^−08^
**Path ID**	**Pathway Names in Cluster Two**	**Found**	**Total**	**Ratio**	** *p* **	**pFDR**
R-HSA-140875	Common pathway of fibrin clot formation	16	25	0.001719	1.11 × 10^−16^	4.33 × 10^−15^
R-HSA-8957275	Post-translational protein phosphorylation	17	109	0.007496	1.11 × 10^−16^	4.33 × 10^−15^
R-HSA-216083	Integrin cell-surface interactions	15	86	0.005914	1.11 × 10^−16^	4.33 × 10^−15^
R-HSA-140877	Formation of fibrin clot (Clotting Cascade)	24	43	0.002957	1.11 × 10^−16^	4.33 × 10^−15^
R-HSA-76009	Platelet aggregation (Plug Formation)	13	53	0.003645	1.11 × 10^−16^	4.33 × 10^−15^
R-HSA-140837	Intrinsic pathway of fibrin clot formation	14	26	0.001788	1.11 × 10^−16^	4.33 × 10^−15^
R-HSA-109582	Hemostasis	49	801	0.055082	1.11 × 10^−16^	4.33 × 10^−15^
R-HSA-114608	Platelet degranulation	24	139	0.009559	1.11 × 10^−16^	4.33 × 10^−15^
R-HSA-76002	Platelet activation, signaling and aggregation	31	291	0.020011	1.11 × 10^−16^	4.33 × 10^−15^
R-HSA-76005	Response to elevated platelet cytosolic Ca^2+^	24	146	0.01004	1.11 × 10^−16^	4.33 × 10^−15^
R-HSA-381426	Regulation of insulin-like growth factor (IGF) transport and uptake by insulin-like growth factor binding proteins (IGFBPs)	19	127	0.008733	1.11 × 10^−16^	4.33 × 10^−15^
R-HSA-1566948	Elastic fiber formation	12	46	0.003163	8.88 × 10^−16^	3.20 × 10^−14^
R-HSA-2129379	Molecules associated with elastic fibers	11	38	0.002613	5.00 × 10^−15^	1.65 × 10^−13^

FDR: False Discovery Rate; TNFR1: tumor necrosis factor receptor 1; TNF: tumor necrosis factor; TRIF(TICAM1): Toll-Like Receptor Adaptor Molecule 1.

**Table 3 ijms-22-12108-t003:** Results of Molecular Complex Detection (MCODE) analysis performed on differentially expressed proteins (DEPs) and genes of death due to ischemic stroke.

MCODE Components	GO ID	Biological Term	Log10 (*p*) Value
All DEPs/genes, cluster one, MCODE2	R-HSA-3000178	ECM proteoglycans	−7.1
	WP306	Focal adhesion	−5.8
	R-HSA-1474244	Extracellular matrix organization	−5.3
All DEPs/genes, cluster two, MCODE1	ko04668	TNF signaling pathway	−40.4
	hsa04668	TNF signaling pathway	−39.8
	ko04657	IL-17 signaling pathway	−28.7

GO ID: Gene Ontology Identification; ECM: extracellular matrix; TNF: tumor necrosis factor: IL: interleukin.

**Table 4 ijms-22-12108-t004:** KEGG pathway classifications of the differently expressed proteins and genes in cluster one and two of death due to stroke.

Path ID	Pathway Names Associated with Cluster One	Observed	Background	Strength	pFDR
hsa04610	Complement and coagulation cascades	27	78	1.89	4.62 × 10^−39^
hsa04510	Focal adhesion	24	197	1.43	4.40 × 10^−25^
hsa05205	Proteoglycans in cancer	23	195	1.42	7.70 × 10^−24^
hsa04151	PI3K-Akt signaling pathway	25	348	1.2	3.09 × 10^−21^
hsa04611	Platelet activation	18	123	1.51	3.93 × 10^−20^
hsa04512	ECM-receptor interaction	16	81	1.64	9.21 × 10^−20^
hsa04933	AGE-RAGE signaling pathway in diabetic complications	16	98	1.56	1.21 × 10^−18^
hsa04926	Relaxin signaling pathway	16	130	1.44	6.38 × 10^−17^
hsa04810	Regulation of actin cytoskeleton	14	205	1.18	9.32 × 10^−12^
hsa04068	FoxO signaling pathway	12	130	1.31	1.52 × 10^−11^
**Path ID**	**Pathway Names Associated with Cluster Two**	**Observed**	**Background**	**Strength**	**pFDR**
hsa04668	TNF signaling pathway	30	108	1.81	4.79 × 10^−42^
hsa04060	Cytokine-cytokine receptor interaction	35	263	1.49	5.35 × 10^−40^
hsa04620	Toll-like receptor signaling pathway	23	102	1.72	3.08 × 10^−30^
hsa04657	IL-17 signaling pathway	22	92	1.75	1.44 × 10^−29^
hsa04064	NF-kappa B signaling pathway	22	93	1.74	1.48 × 10^−29^
hsa04621	NOD-like receptor signaling pathway	25	166	1.55	1.84 × 10^−29^
hsa04630	Jak-STAT signaling pathway	22	160	1.51	3.66 × 10^−25^
hsa04380	Osteoclast differentiation	20	124	1.57	4.30 × 10^−24^
hsa04659	Th17 cell differentiation	19	102	1.64	5.84 × 10^−24^
hsa04622	RIG-I-like receptor signaling pathway	17	70	1.75	3.72 × 10^−23^

KEGG: Kyoto Encyclopedia of Genes and Genomes; ID: Identification; FDR: False Discovery Rate; PI3K-Akt: Phosphatidylinositol 3-kinase-protein kinase B; ECM: extracellular matrix; AGE-RAGE: Advanced glycation end products and their receptors; TNF: tumor necrosis factor: IL: interleukin; NF-kappa B: nuclear factor kappa B; NOD: nucleotide-binding, and oligomerization domain; Jak-STAT: Janus kinase-signal transducer and activator of transcription; Th: T helper; RIG: Retinoic acid-inducible gene.

**Table 5 ijms-22-12108-t005:** Results of inBio Discover annotation analysis with the DOID disease annotations classification in ischemic death due to stroke proteins and genes.

DOID ID	Disease	Size	Overlap	Enrichment	*p*-Value
DOID:0060032	Autoimmune disease of the musculoskeletal system	645	62/267	7.20	1.8 × 10^−35^
DOID:1247	Blood coagulation disease	238	42/267	13.22	7.3 × 10^−35^
DOID:612	Primary immunodeficiency syndrome	1.3k	83/267	4.67	5.0 × 10^−34^
DOID:74	Hematopoietic disease	1.6k	91/267	4.20	5.4 × 10^−34^
DOID:2349	Atherosclerosis	363	48/247	9.90	9.4 × 10^−34^
DOID:7148	Rheumatoid arthritis	313	45/247	10.77	2.9 × 10^−33^
DOID:2348	Atherosclerotic cardiovascular disease	352	47/247	10.00	3.0 × 10^−33^
DOID:417	Autoimmune disease	1.1k	74/246	5.20	3.3 × 10^−33^
DOID:0060903	Thrombosis	108	31/247	21.50	5.9 × 10^−33^
DOID:2941	Immune system disease	1.9k	95/267	3.75	1.2 × 10^−31^

DOID ID: Disease Ontology—Institute for Genome Sciences @ University of Maryland (disease-ontology.org (accessed on 19 September 2021)).

**Table 6 ijms-22-12108-t006:** REACTOME pathways and PANTHER biological processes statistically over-represented in the DEPs/gene list of death due to stroke.

REACTOME Pathways	Total	Expected	Hits	*p*	pFDR
Metabolism of mRNA	317	30.3	95	4.15 × 10^−26^	5.82 × 10^−23^
Metabolism of RNA	339	32.4	98	1.39 × 10^−25^	9.73 × 10^−23^
3′-UTR-mediated translational regulation	201	19.2	65	6.16 × 10^−20^	1.73 × 10^−17^
L13a-mediated translational silencing of ceruloplasmin expression	201	19.2	65	6.16 × 10^−20^	1.73 × 10^−17^
Translation	249	23.8	73	1.63 × 10^−19^	3.81 × 10^−17^
Nonsense-mediated decay independent of the exon junction complex	184	17.6	61	2.27 × 10^−19^	4.54 × 10^−17^
GTP hydrolysis and joining of the 60S ribosomal subunit	201	19.2	64	3.02 × 10^−19^	5.30 × 10^−17^
Eukaryotic translation elongation	186	17.8	61	4.19 × 10^−19^	6.53 × 10^−17^
Nonsense-mediated decay enhanced by the exon junction complex	203	19.4	64	5.39 × 10^−19^	6.73 × 10^−17^
Nonsense-mediated decay	203	19.4	64	5.39 × 10^−19^	6.73 × 10^−17^
**PANTHER biological processes**					
Translation	315	21.2	93	3.74 × 10^−36^	7.25 × 10^−34^
MRNA splicing, via spliceosome	236	15.9	52	1.26 × 10^−14^	8.12 × 10^−13^
RNA splicing	289	19.5	55	1.33 × 10^−12^	6.44 × 10^−11^
RNA metabolic process	47	3.17	20	5.08 × 10^−12^	1.97 × 10^−10^
Protein folding	157	10.6	37	1.09 × 10^−11^	3.52 × 10^−10^
MRNA processing	370	24.9	61	4.62 × 10^−11^	1.28 × 10^−09^
Rhythmic process	124	8.35	31	1.05 × 10^−10^	2.43 × 10^−09^
Blood coagulation	193	13	40	1.13 × 10^−10^	2.43 × 10^−09^
Cell-matrix adhesion	91	6.13	25	8.03 × 10^−10^	1.56 × 10^−08^

FDR: False Discovery Rate; UTR: untranslated region; GTP: Guanosine triphosphate.

## Data Availability

The data presented in this study are openly available in the article.

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
