# Peer review of "New Drug Targets to Prevent Death Due to Stroke: A Review Based on Results of Protein-Protein Interaction Network, Enrichment, and Annotation Analyses"

_ijms, 2021, doi:10.3390/ijms222212108_

Round 1

Reviewer 1 Report

Acute ischemic stroke is one of the leading causes of death and disability worldwide.  The novelty of this study is the discovery of new putative drug targets in signalling pathways, cellular components, molecular patterns, and transcription factors enriched in protein-protein interaction networks in IS -associated mortality. The authors have created three types of networks, the first - DEP - for their own research, the second - for the combination DEP + genes; and the third for all data together. The discussion chapter is well structured and describes in detail the involvement of PPI, DEP/genes in clusters. Below are some comments on the article: 1. the introduction is too long, it needs to be shortened; 2. on the basis of what parameters did the authors decide that ALB and VWF are switched between 1 and 2 clusters? The author should explain. 3. why KEGG enrichment analysis was performed for the first protein subnetwork and not for the second protein subnetwork (Figure 3), GO Molecular Function (Figure 5) for the second cluster? InterPro domain analysis is presented for 2 clusters (Figure 4 and Figure 6).  This should be explained by the author. I recommend combining the enrichment images (KEGG, InterPro, GO) for each cluster separately to better identify the material. 4. check the numbering of the tables (2 tables numbered 5); the numbering of the subheadings (3.1. The networks and subnetworks of death by stroke - 3.1. terms and functions overrepresented in the hemostasis subnetwork); 5. abbreviations should be introduced at the first mention. If you have entered abbreviations, for example: ischemic stroke (IS).

Author Response

"please see attachment"

Reviewer 2 Report

An excellent and informative paper. Authors are to be commended. Information is well presented in a clear and easy to follow style. Data are well interrogated and interpreted using the appropriate approaches. I believe the paper to be of interest to a wide readership with the potential of high citations. Only modification is minor proof reading eg- ALB in Figure 1 is hidden. Bring to front.

Author Response

"please see the attachment"

Reviewer 3 Report

In this review, Maes et al., conducted a variety of secondary data analysis on data sets of ischemic stroke patients to identify various physiological markers which are correlated with patient death. Although the identified immune and homeostatic markers have been widely reported, this review reinforces and supplements existing analyses literature.

A few major issues should be addressed to strengthen this manuscript:

Foremost, specific datasets which provided the inputs for manuscript analysis should be included in methods. Given the differential population and information listed in data library, it would be helpful to provide general patient characteristics of input data; as authors mentioned in Discussion of the differential expression of certain genes across various populations.

General timing and methods of sample collection should be described in brief in Methods.

If possible, analysis should discriminate between those patients receiving tPA vs those with only symptomatic treatment. This is an especially major point as tPA administration skews the expression of not only the homeostatic genes identified but also those of the immune subnetwork. Moreover, the issue of tPA resistance where approx. 33% of patients receiving the thrombolytic achieves complete or partial recanalization;, the remaining tPA-treated patients without recanalization experience worsened outcome/death in an unknown process in which immune and endothelial factors contribute substantially. This again would skew manuscript findings. If this cannot be accounted for with datasets, should be listed in limitations.

Lastly, this review should tone down their broad claims in Conclusions. Of course it is easy to say that these major pathways should be targeted, but if so then authors should also mention the myriad RCTs targeting major pathways highlighted and why few reach endpoint. These secondary analyses reinforce what is already known and reported previously, major gaps in timing and perspective on combination/singular targeting remain.

Author Response

REFEREE 3.

In this review, Maes et al., conducted a variety of secondary data analysis on data sets of ischemic stroke patients to identify various physiological markers which are correlated with patient death. Although the identified immune and homeostatic markers have been widely reported, this review reinforces and supplements existing analyses literature.

A few major issues should be addressed to strengthen this manuscript:

Foremost, specific datasets which provided the inputs for manuscript analysis should be included in methods. Given the differential population and information listed in data library, it would be helpful to provide general patient characteristics of input data; as authors mentioned in Discussion of the differential expression of certain genes across various populations.

General timing and methods of sample collection should be described in brief in Methods.

@ANSWER: We added ESF2 Table 1 summarizing the charactersitics of all studies included (with age, seks ratio, outcome data, tPA treatment and follow up period.

If possible, analysis should discriminate between those patients receiving tPA vs those with only symptomatic treatment. This is an especially major point as tPA administration skews the expression of not only the homeostatic genes identified but also those of the immune subnetwork. Moreover, the issue of tPA resistance where approx. 33% of patients receiving the thrombolytic achieves complete or partial recanalization;, the remaining tPA-treated patients without recanalization experience worsened outcome/death in an unknown process in which immune and endothelial factors contribute substantially. This again would skew manuscript findings. If this cannot be accounted for with datasets, should be listed in limitations.

@@ANSWER: None of the subjects in the DEP studies were treated witn tPA. So, the results of the biomarkers and outcome data are not affected by tPA. So, the findings are not skewed at all. They are representative of death in a cohort without tPA treatment. In the gene studies per definition tPA will not affect the markers, but an effect on the outcome variable is always possible. This is addressed in the text as:

Additionally, none of the subjects in our DEP studies received tPA, which could have influenced biomarker levels and outcome data. AS such, our DEP results are representative of the pathways leading to death in a cohort of IS patients who did not receive tPA treatment. In the second section of our studies, we included gene studies, and only one of these adjusted the results for tPA treatment while another was performed on stroke subjects who were administered tPA. Most studies, however, did not offer any information concerning tPA treatment. Therefore, future research should focus on the PPI network associated with death from IS in patients treated with tPA in order to delineate drug targets that may augment the efficacy of tPA in preventing death from IS.

Lastly, this review should tone down their broad claims in Conclusions. Of course it is easy to say that these major pathways should be targeted, but if so then authors should also mention the myriad RCTs targeting major pathways highlighted and why few reach endpoint. These secondary analyses reinforce what is already known and reported previously, major gaps in timing and perspective on combination/singular targeting remain.

@@ANSWER: We now discuss the pro anc contras of targeting the pathways and molecular functions delineated in our study. In addition we have deleted teh last three lines of the discussion. The added setions reads:

 Nevertheless, administration of VEGF and increased endogenous VEGF signaling may have negative consequences in acute stroke due to BBB breakdown causing neuroinflammation and intracranial pressure [158]. There are some studies which showed that TGF-β1 has a neuroprotective effect against ischemia-associated neuronal injuries and excitotoxicity [159]. Preclinical research shows that some proteoglycans (mimetics or recombinant proteoglycan cores) may be attractive drugs for targeting (neuro)degenerative disorders, including stroke [160]. For example, in a rodent model, heparan sulfate proteoglycan 2 shows improved angiogenesis after IS and is neuroprotective [161].

And

Recently, Zhu et al. [162] detected that, in animal models, Janus Kinase inhibition may improve IS injury and neuroinflammation. Nevertheless, there are many contradictory results of targeting Janus Kinases in IS models as reviewed by Raible et al. [163]. A recent review showed that targeting the NLRP3 inflammasome may offer another strategy to improve IS [164]. The TLRs and especially TLR2/4 play a critical role in IS, the pathological development of IS brain damage, and reperfusion injury [165, 166]. Inhibition of the TLR4-NOX4 pathway, a predominant causal pathway in IS, protects against oxidative stress and neuronal apoptosis [167]. In addition, preconditioning of the TLR response to damage associated patterns may be associated with increased neuroprotection [168].

Round 2

Reviewer 3 Report

Author response and revisions improves manuscript.